# Discovering functional sequences with RELICS, an analysis method for CRISPR screens

**Patrick C. Fiaux**[1,2]*, **Hsiuyi V. Chen**[2], **Poshen B. Chen**[3], **Aaron R. Chen**[2,4], **Graham McVicker**[2]*

**1** Bioinformatics and Systems Biology Graduate Program, University of California, San Diego, La Jolla, CA, Unites States of America, **2** Integrative Biology Laboratory, Salk Institute for Biological Studies, La Jolla, CA, Unites States of America, **3** Ludwig Institute for Cancer Research, La Jolla, CA, Unites States of America, **4** Division of Biological Sciences, University of California, San Diego, La Jolla, CA, Unites States of America

* pfiaux@ucsd.edu (PCF); gmcvicker@salk.edu (GMV)

## Abstract

CRISPR screens are a powerful technology for the identification of genome sequences that affect cellular phenotypes such as gene expression, survival, and proliferation. By targeting non-coding sequences for perturbation, CRISPR screens have the potential to systematically discover novel functional sequences, however, a lack of purpose-built analysis tools limits the effectiveness of this approach. Here we describe RELICS, a Bayesian hierarchical model for the discovery of functional sequences from CRISPR screens. RELICS specifically addresses many of the challenges of non-coding CRISPR screens such as the unknown locations of functional sequences, overdispersion in the observed single guide RNA counts, and the need to combine information across multiple pools in an experiment. RELICS outperforms existing methods with higher precision, higher recall, and finer-resolution predictions on simulated datasets. We apply RELICS to published CRISPR interference and CRISPR activation screens to predict and experimentally validate novel regulatory sequences that are missed by other analysis methods. In summary, RELICS is a powerful new analysis method for CRISPR screens that enables the discovery of functional sequences with unprecedented resolution and accuracy.

## Author summary

Non-coding genome sequences contain a disproportionate number of genetic variants associated with human traits and diseases, however, interpretation of non-coding genetic variants is difficult because the molecular function of most non-coding sequences is unknown. By perturbing the genome using CRISPR, the function of non-coding sequences can be tested. Here we develop a new computational tool, RELICS, for the analysis of high-throughput CRISPR screens in which thousands of genome sequences are perturbed in a single experiment. Using simulated data, we find that RELICS has higher accuracy and resolution than other analysis methods. We apply RELICS to existing datasets to discover novel functional sequences and verify these predictions with experiments.

ⓞ OPEN ACCESS

**Data Availability Statement:** All data were obtained from published papers or simulated. The formatted data that we used for analyses can be

downloaded from figshare: https://figshare.com/projects/RELICS_2_data/74376.

**Funding:** This research was supported by NIH/NIAID grant 2R01AI107027-06; by NIH/NIDDK grant 1 R01 DK122607-01; by the National Cancer Institute funded Salk Institute Cancer Center (NIH/NCI CCSG: 2 P30 014195); by a gift from the Jacobs Foundation; by a fellowship from the H.A. and Mary K. Chapman Charitable Trust to P.C.F.; by a fellowship from the Jesse and Caryl Philips Foundation to P.C.F; by a Salk Alumni Fellowship to H.V.C; and by the Frederick B. Rentschler Developmental Chair to G.M. The funders had no role in study design, data collection and analysis, decision to publish, or preparation of the manuscript.

**Competing interests:** The authors have declared that no competing interests exist.

In summary, RELICS is a powerful new analysis method for the discovery of functional sequences from CRISPR screens.

This is a *PLOS Computational Biology* Methods paper.

## Introduction

CRISPR screens perturb the genome to discover sequences that affect cellular phenotypes such as growth, survival, or gene expression. The first CRISPR screens targeted protein-coding genes [1], however recent advances have enabled screens that tile sgRNAs across non-coding regions of the genome [2–12], allowing the systematic discovery of novel functional sequences in their native genomic context.

In a pooled CRISPR screen, thousands of single-guide RNAs (sgRNAs) are delivered to cells and target sequences for genomic perturbations, which can include mutation, repression (CRISPR interference, CRISPRi), or activation (CRISPRa). To target sequences for mutation, sgRNAs are introduced to cells alongside Cas9. Cas9 creates double strand breaks at the targeted sites, and mutations are introduced by error-prone non-homologous end-joining [13]. In a CRISPRi experiment, targeted sites are silenced by a deactivated Cas9 (dCas9) fused to a repressive domain such as the Krüppel-associated box (dCas9:KRAB) [14, 15]. Similarly, in a CRISPRa experiment, dCas9 is fused to an activation domain such as VP64 or p300 [16–18].

Following genomic perturbation, the cells are subsequently sorted into pools based on a cellular phenotype (e.g. high vs. low gene expression, survival, proliferation), and the distributions of sequenced sgRNAs are compared across pools to identify functional genomic sequences that affect the cellular phenotype used for sorting. For example, sgRNAs that disrupt activating regulatory sequences would result in reduced target gene expression and thus be enriched in pools selected for low target gene expression.

The analysis of tiling CRISPR screens poses numerous challenges including (i) the need to combine information across multiple sequencing pools, (ii) the noisy and overdispersed nature of genomic count data [19, 20], (iii) the spatial organization of sgRNA target sites and functional sequences, and (iv) the 'area of effect' of genomic perturbations, which often extends well beyond the genomic location targeted by the sgRNA (e.g. in CRISPRa or CRISPRi screens activating or silencing epigenetic modifications can spread over 1kb or more from target sites [15]). Currently, no existing methods address all of these challenges and moreover, almost all analysis methods for CRISPR screens were designed for gene-based screens, as opposed to tiling non-coding screens where the identity of functional sequences are unknown.

Here we describe RELICS (**R**egulatory **E**lement **L**ocation **I**dentification in **C**RISPR **S**creens), a new method for the analysis of CRISPR screens which specifically addresses all of the above challenges. RELICS uses a flexible Bayesian hierarchical model to jointly analyze sgRNA counts across multiple pools, uses count distributions that can accommodate overdispersion, and considers the overlapping effects of multiple sgRNAs and the potential presence of multiple nearby functional sequences. RELICS also estimates the total number of functional sequences that are supported by the data. Using simulated data, we demonstrate that RELICS out-performs existing analysis methods (MAGeCK [21], BAGEL [22], and CRISPR-SURF [23]), with better precision and recall across a wide variety of conditions. In addition, the sequences predicted by RELICS have higher resolution—they are smaller but still contain the true functional sequences. Finally, when applied to published datasets, RELICS identifies novel regulatory sequences that we experimentally validate, and that are missed by other analysis methods.

## Results

### The RELICS model

RELICS is designed to discover functional sequences from pooled CRISPR screens including those based on cell survival, cell proliferation, and gene expression (Fig 1A and 1B). RELICS aims to determine both the number and location of functional sequences in the screened genome sequence and includes several important features: (i) it increases power by jointly modeling data from multiple pools; (ii) it models overdispersed sgRNA counts appropriately without requiring transformation of the data or assuming normality; (iii) it considers the spatial organization of sgRNA target sites and functional sequences; (iv) it accounts for the 'area of effect' of sgRNAs; and (v) it provides interpretable probabilities for the genome location of each predicted functional sequence.

One of RELICS' features is its ability to jointly analyze sgRNA count data across multiple pools, while controlling for extra variability (overdispersion) in the data. Modeling genomic count data while accounting for overdispersion increases power and reduces false positives [19, 20, 24]. In RELICS this is accomplished using a Dirichlet multinomial distribution, which describes the probability that a cell containing an sgRNA will be observed in each pool.

It is also important to consider the genome locations of both functional sequences and sgRNA target sites as well as the area of effect for each sgRNA. The area of effect of CRISPR

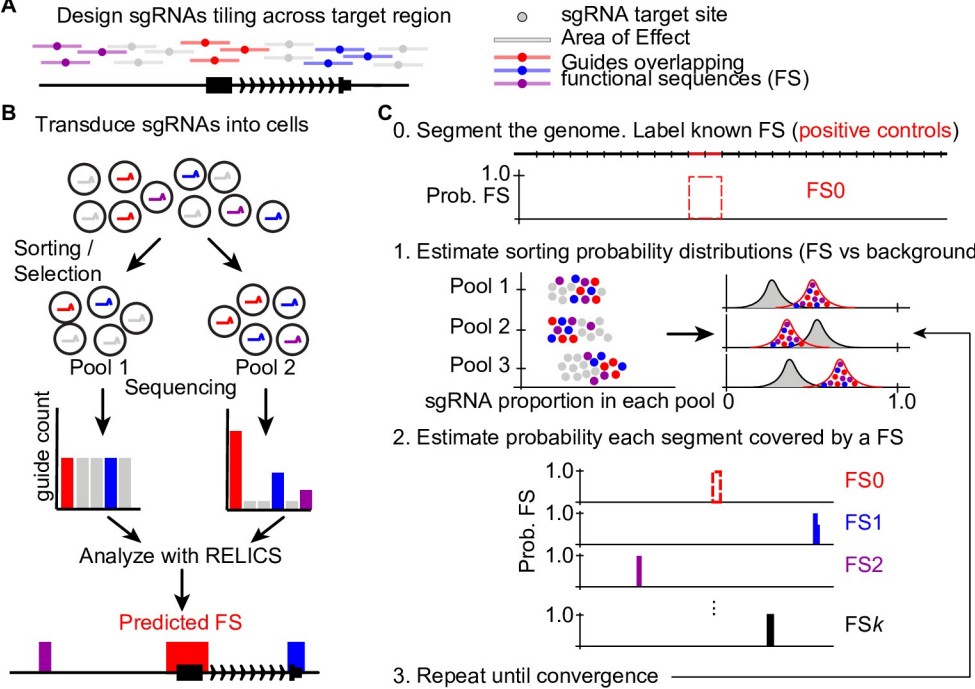

**Fig 1. Schematic of a tiling CRISPR screen and RELICS.** (A) sgRNAs are designed to target sites tiling across a genomic region. Bars around each target site indicate the area of effect. sgRNAs 'overlap' genome sequences within their area of effect. In this example, sgRNAs overlapping functional sequences (FSs) are colored red, purple and blue. (B) In a CRISPR screen, sgRNAs are introduced into cells and the cells are sorted into pools or subjected to selection (e.g. for proliferation or survival). The sgRNAs in each pool are counted and analyzed by RELICS to predict FSs. (C) RELICS predicts FSs using an Iterative Bayesian Stepwise Selection Algorithm. Step 0 (initialization): the screened region is divided into small (e.g. 100bp) genome segments; segments containing known FSs are labeled FS0. Step 1: Using the predicted FSs, RELICS estimates sorting probability distributions for sgRNAs that do/do-not overlap FSs. Step 2: RELICS uses the sorting probability distributions to compute the probability each genome segment contains a functional sequence. Step 3: Steps 1–3 are repeated until convergence.

perturbations means that a single sgRNA can potentially affect multiple functional sequences in the vicinity of the target site. Similarly, a single functional sequence can be perturbed by multiple sgRNAs with nearby target sites. To model the spatial organization of functional sequences and sgRNA-directed perturbations, RELICS divides the sequence targeted by the screen into small windows called genome segments. Each genome segment is then associated with all sgRNAs that are predicted to affect that genome segment, based on their overlap with the sgRNA's area of effect (Fig 1).

RELICS assumes that the counts of sgRNAs are affected when the genome segment they are associated with contains a functional sequence. sgRNAs that affect functional sequences are expected to have a different count distribution across pools, and RELICS uses different Dirichlet distributions for sgRNAs that do or do not overlap functional sequences (Fig 1C). The hyperparameters for both distributions are estimated empirically by maximum likelihood in a semi-supervised manner. Known functional sequences (positive controls), such as genome segments overlapping the promoter or established enhancer of the target gene are used to start this process. We refer to these positive controls as functional sequence 0 (FS0). The hyperparameters can be estimated not only from FS0, but can also be continually updated as new functional sequences (FS1 through FSK) are identified.

Using the hyperparameters and the observed sgRNA counts, RELICS calculates the posterior probability that a genome segment contains a functional sequence. We refer to the probability that a genome segment contains a functional sequence as the 'functional sequence probability'.

Exact calculation of functional sequence probabilities for every genome segment is impractical, however, because all combinations of functional sequence placements would have to be calculated. To overcome this problem, RELICS uses a novel Iterative Bayesian Stepwise Selection (IBSS) algorithm [25] to calculate approximate functional sequence probabilities, while automatically estimating the number of functional sequences, $K$, from the data.

The final output from RELICS is an estimate of $K$, the number of functional sequences, and the probability that each genome segment contains each functional sequence. An important feature of RELICS is that it outputs separate probabilities for each functional sequence, providing discrete genome segments that can be used for follow-up validation experiments (Fig 1C). We plot these functional sequence probabilities as separate tracks or as a combined track with different colors. In addition, the functional sequences output by RELICS are rank-ordered, with functional sequence 1 (FS1) having the strongest statistical support.

## Application of RELICS to simulated data

To evaluate the performance of RELICS and other analysis methods, we developed a CRISPR screen simulation framework, CRSsim, that generates realistic datasets where the ground truth is known (S1 Fig). We compared the simulated data to experimental data and found it to be highly similar (S1 Fig). The simulated datasets are useful to benchmark performance since there is currently no large gold-standard set of known functional sequences. We simulated datasets with two different sequencing depths (medium/high), three sgRNA efficiency distributions (low/medium/high), and two selection strengths (weak/strong). The selection strength describes the probability of a cell being sorted into each pool given the presence or absence of an sgRNA targeting a functional sequence. We performed 30 simulations for the 12 combinations of parameter settings under two experimental scenarios: scenario 1 a cellular proliferation screen with two pools per replicate, and scenario 2 a gene expression screen with four pools per replicate. In total we performed 720 simulations, with each simulation consisting of 8,700 sgRNAs targeting a 150kb region with 8 functional sequences each spanning 50bp.

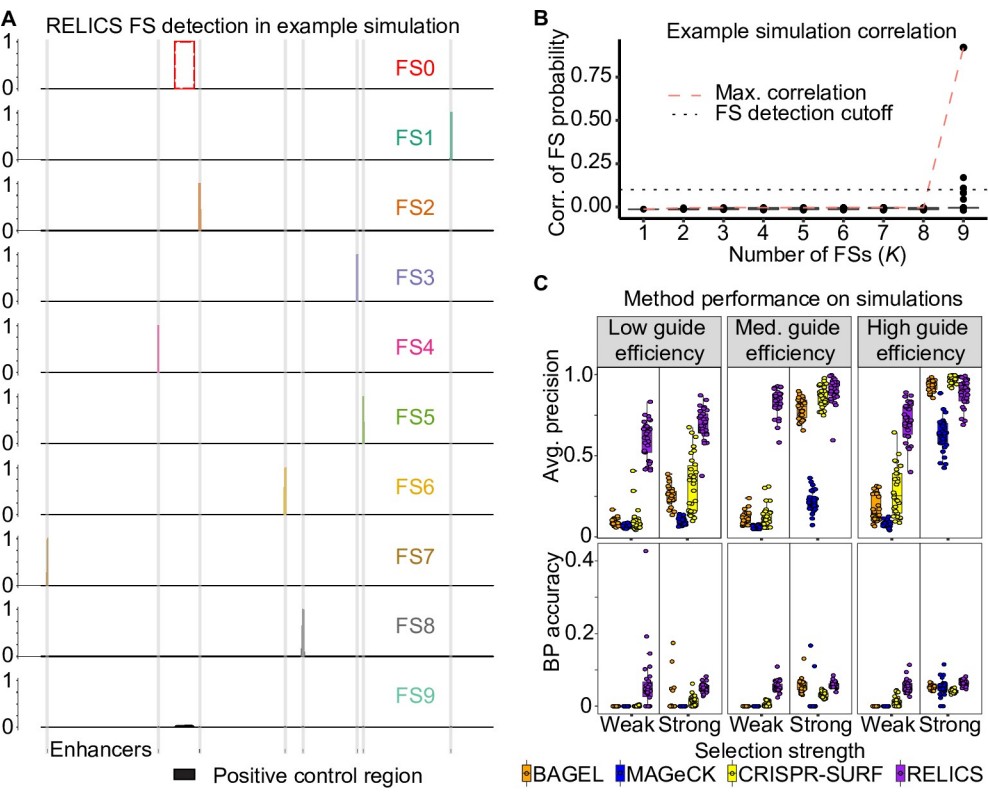

**Fig 2. RELICS functional sequence (FS) prediction and performance of RELICS and other analysis methods on simulated data.** (A) Example of functional sequence (FS) probabilities output by RELICS for a single simulation. FS0 is the known positive control sequence. RELICS correctly predicts all 8 of the simulated enhancers (grey vertical bars), which it labels FS1-FS8. FS9 is displayed to demonstrate the choice of $K = 8$. (B) Boxplots of pairwise correlations of FS probabilities for different choices of $K$ (number of FSs). The red dashed line is the maximum pairwise correlation. The black dotted line is the cutoff used to determine $K$. In this example, RELICS chooses $K = 8$. The hinges of the boxplots correspond to the first and third quartiles, the center lines are the medians, and the whiskers extend to the furthest datapoints that are within 1.5x the interquartile range from the hinge. (C) Performance on 180 simulations summarized by average precision and base pair (BP) accuracy. For this figure, 30 simulations were performed for 6 simulation scenarios with high read depth and different values of the key parameters "selection strength" and "guide efficiency". Additional simulations and performance metrics are provided in S2 Fig, S3 Fig, and S4 Fig. BP accuracy is the fraction of the significant bases identified by each method which are true functional sequences. In these simulations, all methods predict regions that are substantially larger than the small (50bp) simulated functional sequences due to the comparatively large area of effect around sgRNA target sites (1000bp), RELICS predicts the smallest regions with the highest BP accuracy.

We ran RELICS on the simulated datasets and allowed it to estimate the number of functional sequences, $K$, by calculating the pairwise correlation between functional sequence probabilities for different values of $K$. An example of RELICS' predictions on a single simulated dataset is shown in Fig 2. In this example, RELICS identified all 8 of the functional sequences before the pairwise correlations between functional sequence probabilities increased due to overlapping placements (Fig 2A and 2B).

We compared RELICS' predictions on simulated data to those from (i) MAGeCK [21], the dominant method for gene-based CRISPR screens; (ii) BAGEL [22], a supervised method for gene-based CRISPR screens; and (iii) CRISPR-SURF [23], a new tool for tiling CRISPR screens. While both MAGeCK and BAGEL are designed for gene-based screens, we compare against them because MAGeCK is the most popular analysis method for CRISPR screens and has previously been used to analyze tiling CRISPR screens [11]. We use BAGEL to provide a comparison to another method that utilizes training data to estimate model parameters.

We first quantified performance using scores output by each method and computing the average precision (AP), which summarizes a precision recall curve. As scores, we used the functional sequence probability (RELICS), the effect size (CRISPR-SURF), the BayesFactor (BAGEL), and the -log10(FDR) (MAGeCK). We found that RELICS has by far the best performance with the highest AP in 321/360 simulations for scenario 1 and 276/360 simulations for scenario 2 (Fig 2C, S2A Fig, S3A Fig). We next quantified performance using the functional sequences that each method reported as significant. RELICS had the highest precision in 338/360 of the simulations in scenario 1 and 324/360 of the simulations in scenario 2 (S2B Fig, S3B Fig). RELICS also had the highest recall in 331/360 of the scenario 1 simulations and 359/360 of the scenario 2 simulations (S2C Fig, S3C Fig). Notably, RELICS has far better performance than the other methods in scenarios with low sgRNA efficiency and/or weak selection strength. This is important because many sgRNAs in screening libraries have medium or low efficiency [26].

We quantified the resolution of predicted functional sequences with basepair (BP) accuracy —the total number of significant genomic bases that overlap with true simulated functional sequences. For each simulated functional sequence, we quantified the fraction of base pairs that were called significant and overlapped a simulated functional sequence. Due to the large area of effect (1000bp) and small functional sequences (50bp) used by the simulations, all of the methods had low BP accuracy and predicted regions that are substantially larger than the functional sequences. Nonetheless, the predictions made by RELICS covered much smaller regions than the other methods and it had the highest BP accuracy in 313/360 of the scenario 1 simulations and 304/360 of the scenario 2 simulations (Fig 2C, S2D Fig, S3D Fig).

We also simulated data using a tool recently developed by Bodapati et al. [27]. Using the data from these simulations RELICS again outperformed all other methods in AP, precision, recall and BP accuracy (see methods, S4 Fig). Note that neither of the simulations are biased to favor RELICS since CRSsim was designed to mimic experimental procedures and is not tailored to RELICS, and the Bodapati et al. simulation tool was developed by an independent research group. In summary, RELICS outperforms the other methods on simulated data and has the best overall precision, recall, and resolution.

## Application of RELICS to published datasets

We applied RELICS to four published tiling CRISPR screens. The first two data sets are CRISPRa screens for functional sequences that affect the expression of *CD69* and *IL2RA* in Jurkat T cells [9]. For both genes, cells were flow sorted into four pools based on expression (negative, low, medium, high), and in the published analysis putative functional sequences were identified by computing log fold change between pairs of pools. We ran RELICS, CRISPR-SURF, and MAGeCK on both datasets. RELICS predicts $K = 5$ functional sequences for *CD69*, which we label as FS1 through FS5 (Fig 3, S5 Fig). In contrast, MAGeCK predicts a large number of significant regions (23) under a false discovery rate of 0.05, while CRISPR-SURF predicts multiple large functional sequences at and downstream of the *CD69* promoter (Fig 3B).

To test a subset of the *CD69* functional sequence predictions, we designed sgRNAs to target the putative functional sequences, cloned the sgRNAs into lentiviral vectors, and transduced them into Jurkat T cells expressing dCas9:VP64. As a negative control we used a non-targeting sgRNA, and as positive control we used an sgRNA targeting a site near the *CD69* transcription start site (Fig 3C and 3D). The region including FS1 and FS2 was previously found to activate *CD69* expression [9] and we confirm that CRISPRa targeting of this region increased the number of *CD69* positive cells. Similarly, we validated that FS4 also increases *CD69* expression. All methods detected a signal at FS3, however we were not able to confirm that targeting of this

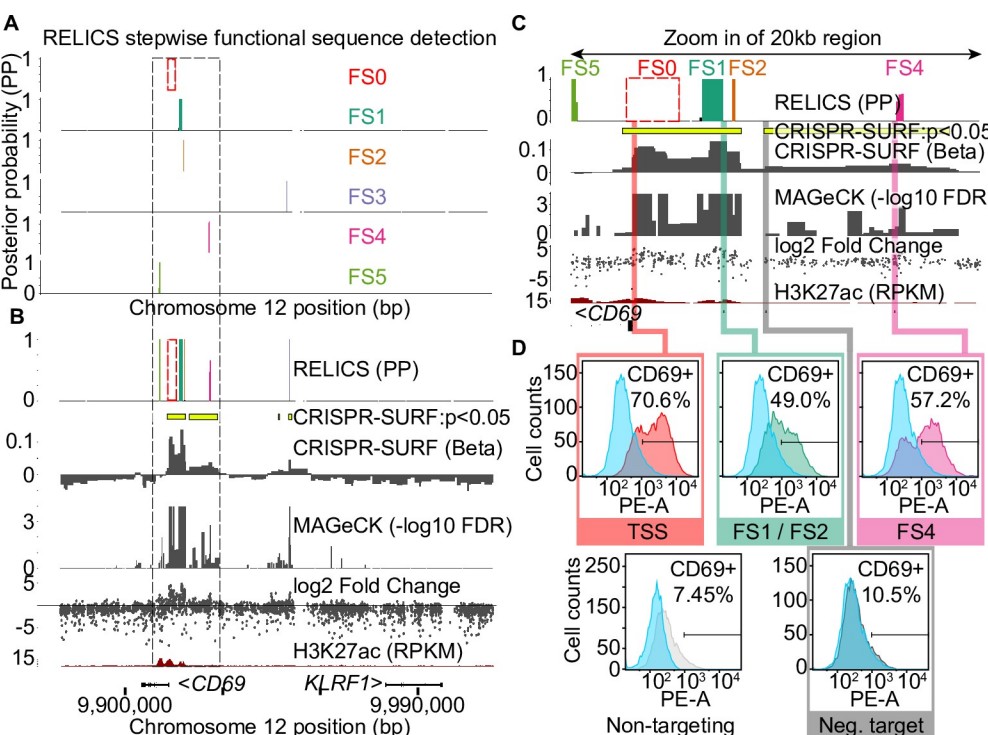

**Fig 3. Analysis of a published CRISPR activation (CRISPRa) screen for *CD69* expression in Jurkat T cells.** (A) RELICS detects 5 functional sequences (FSs) labeled FS1-FS5. FS0 is a known positive control sequence (the CD69 promoter) provided as input to RELICS and CRISPR-SURF. (B) Analysis of the CD69 screen by RELICS, CRISPR-SURF, MAGeCK and log2 fold change. The RELICS probabilities for each FS are collapsed into a single track. An H3K27ac ChIP-seq track for Jurkat cells is included. (C) Zoom in of a 20kb region (indicated by dashed box in (a) and (b)). Experimentally tested regions are indicated by colored bars. (D) Experimental validations. Lentiviruses carrying sgRNAs targeting different regions were transduced to Jurkat cells expressing dCas9:VP64, and the expression of CD69 was measured by flow cytometry using PE-conjugated anti-human CD69 antibody. The results from each experiment are overlaid those from a non-targeting negative control sgRNA (blue). Targeting sgRNAs were chosen for their high specificity and high predicted efficiency (relative to possible sgRNAs in the region) and in some cases are adjacent to the predicted FS rather than within the FS. Results from additional validation experiments are shown in S6 Fig.

region affects *CD69* expression (S6 Fig). Finally, targeting a "negative sequence" that was not predicted by RELICS, but is on the edge of a large significant region reported by CRISPR-SURF, did not change *CD69* expression. These results confirm that the predictions made by RELICS are accurate. In addition, the output from RELICS is easier to interpret because it provides ranked, cleanly-delineated predictions for each functional sequence (Fig 3A).

RELICS identified 15 functional sequences for *IL2RA*, of which all but one (FS6) are within or very close to the 6 regions identified in the original study (S7 Fig). Since RELICS predictions have higher resolution than other methods, the multiple smaller regions predicted by RELICS may reflect the true presence of multiple functional sequences. Alternatively, some large functional sequences may be split into smaller sequences by RELICS, particularly if some of the sgRNAs targeting the middle of the sequence have very low efficiency.

We next applied RELICS to a CRISPRi proliferation screen surrounding the *MYC* locus in K562 cells [3] (Fig 4). As with *IL2RA*, RELICS detected all of the previously-reported signals, but split one of them into smaller predicted functional sequences. Interestingly, RELICS also identified 6 regions that have not been previously reported and that neither CRISPR-SURF nor MAGeCK detected. To test whether these sequences are functional we targeted 5 of them

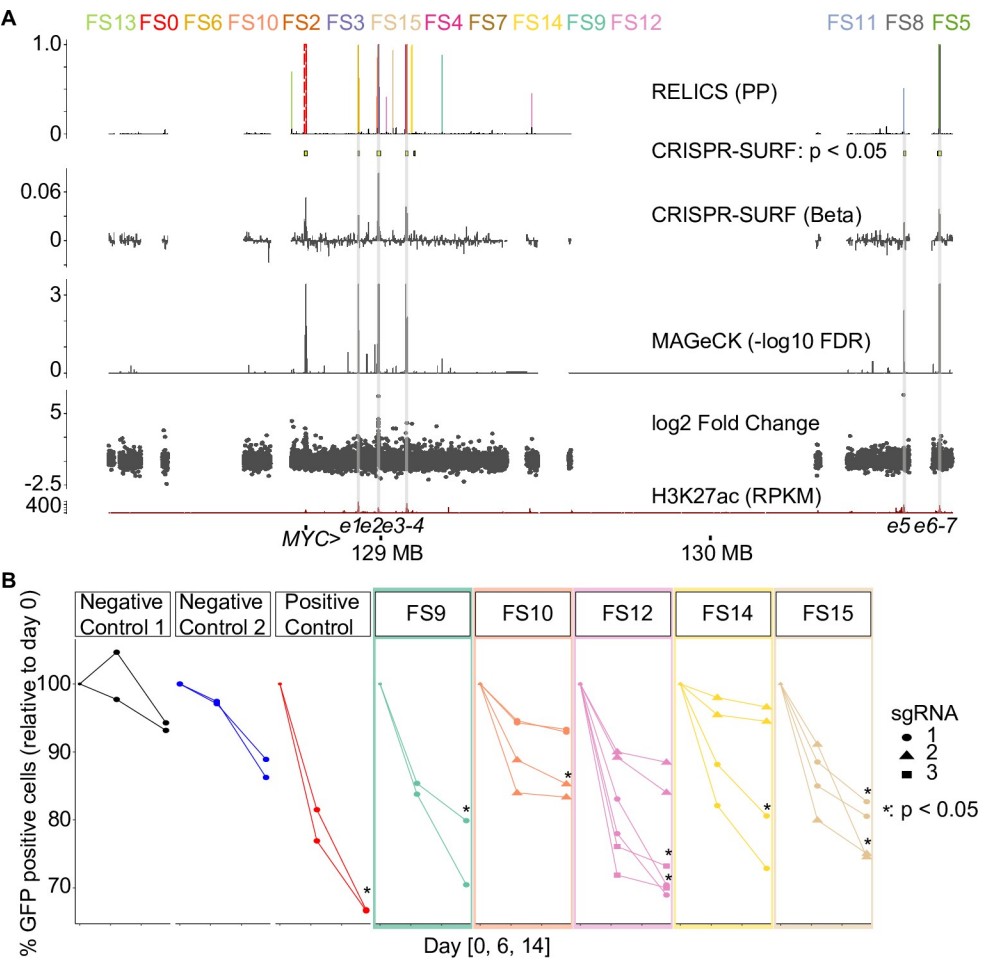

**Fig 4. Analysis of a *MYC* CRISPRi cellular proliferation screen published by Fulco et al. 2016.** (A) Output of RELICS and other analysis methods. Each FS predicted by RELICS is given a different color and the labels are arranged by genomic position. (B) Experimental validations. Each validation experiment is a cellular proliferation assay, in which the percent of GFP-positive cells (i.e. those that received the sgRNA) are measured at day 0, day 6 and day 14. As negative controls we used dCas9:KRAB only (no sgRNA) as well as sgRNAs targeting a 'safe harbor' non-functional region on chromosome 8. As positive controls we used sgRNAs targeting a known regulatory region (e2, identified by Fulco et al., corresponding to FS2 and FS3). Each functional sequence was targeted with either one, two, or three different sgRNAs with two replicates each. sgRNAs that resulted in a significant reduction in % GFP compared to negative controls (dCas9 only, Neg. Ctrl.) at day 14 have an asterisk (Student's one-sided t-test, $p < 0.05$).

with CRISPRi. As a positive control we targeted a previously-detected functional sequence reported by Fulco et al. (referred to as e2), and as negative controls we used dCas9:KRAB alone or dCas9:KRAB with an sgRNA targeting a safe harbor sequence on chromosome 8 with no known function. Of the 5 FSs we tested, 4/5 showed a substantial decrease in proliferation and 1/5 (FS10) showed a small decrease in proliferation with at least one of the targeting sgRNAs (Fig 4B). This confirms that RELICS discovered additional functional sequences that are missed by other methods.

Finally, we applied RELICS to a CRISPRi proliferation screen surrounding *GATA1* [3]. RELICS predicted five functional sequences, two of which (FS1 and FS2) have been previously validated [3], two of which are within *GATA1* (FS3, FS4), and one of which is upstream of *GATA1* (FS5) but not detected by any of the other methods (Fig 5A). We confirmed that inhibition of this newly-discovered functional sequence with CRISPRi reduces proliferation (Fig

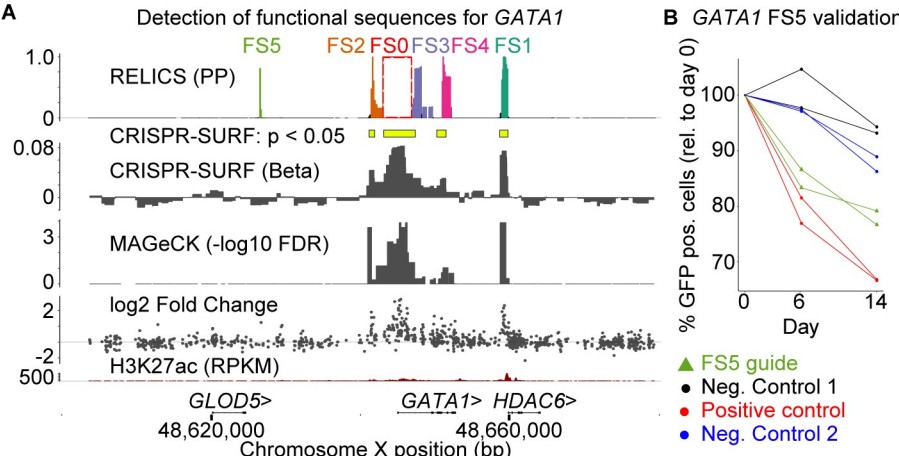

**Fig 5. Analysis of a published CRISPR inhibition (CRISPRi) cellular proliferation screen around *GATA1*.** (A) Results from RELICS and other analysis methods. RELICS detects 5 functional sequences (FS1-5). FS1 and FS2 have previously been validated; FS3 and FS4 fall within GATA1. (B) Results from validation experiments (2 replicates each). As negative controls we used dCas9:KRAB only as well as guides targeting a non-functional region on chromosome 8. As positive controls we used sgRNA targeting a known regulatory region (e2, previously identified by Fulco et al. 2016). Each validation experiment is a cellular proliferation assay, in which the percent of GFP-positive cells (i.e. those that received the sgRNA) are measured at day 0, day 6 and day 14. Targeting the positive control region and FS5 clearly reduces proliferation compared to the negative controls.

5B). The original study also predicted a functional region near *GLOD5* (S8A Fig). However, neither RELICS nor the other two methods identified a functional sequence at this location. This suggested to us that the region near *GLOD5* may have been a false positive detected in the original study. To test this hypothesis, we used CRISPRi to target the *GLOD5* region, the *GATA1* promoter (positive control), and a sequence on chromosome 8 (negative control). While the sgRNAs targeting the *GATA1* promoter decreased cellular proliferation as expected, targeting of the *GLOD5* region did not affect proliferation relative to the negative control (S8B Fig). Thus, this region is unlikely to be a functional sequence, and the low probability reported by RELICS is appropriate. Overall, RELICS accurately identifies functional sequences, and provides an easily interpretable output that quantifies its certainty that a region contains a functional sequence.

## Discussion

We have developed RELICS, a Bayesian hierarchical model for the analysis of CRISPR screens. Unlike gene-based analysis methods, RELICS is specifically designed to analyze CRISPR screens where the locations of functional sequences are not known. The RELICS model provides numerous advantages. First, it considers the overlapping effects of multiple nearby sgRNA target sites and functional sequences. Second, it provides interpretable probabilistic output for each functional sequence that can be used to delineate small genome regions that confidently contain each functional sequence. Third, it models sgRNA counts appropriately without requiring transformation of the data or assuming normality. Fourth, it increases power by jointly modeling data from multiple sgRNA pools. While other methods include some of these features (e.g. CRISPR-SURF deconvolves the effects of multiple sgRNAs, and MAGeCK models overdispersed count data with a negative binomial distribution), only REL-ICS combines all of them into a single model. We also tested the robustness of RELICS to sequencing depth and density of sgRNA target sites by downsampling 75% of the sgRNA

counts as well as 75% of all sgRNAs for the *CD69* CRISPRa screen (S9 Fig). In both cases, the RELICS predictions remained consistent indicating that it is relatively insensitive to sequencing depth and sgRNA density. This is also supported by our simulations, which found that sgRNA efficiency and functional sequence strength affect performance more than sequencing depth (S2 Fig, S3 Fig).

To test the performance of RELICS and other analysis methods we developed a simulation tool, CRSsim. Using CRSsim, we generated hundreds of datasets, performed the first systematic comparison of analysis methods for non-coding CRISPR screens, and found that RELICS has the best performance under a wide variety of conditions. CRSsim is open-source and we envision that it will be a useful tool for future performance comparisons, power analyses, and making informed decisions about experimental designs for CRISPR screens (e.g. spacing of sites targeted by sgRNAs, the sequencing depth, and the complexity of the vector library).

RELICS leverages known functional sequences (labeled positive controls) as well as unlabeled sequences to learn model hyperparameters. It is therefore a semi-supervised method. CRISPR-SURF and BAGEL similarly make use of known control sequences. A strength of this approach is that RELICS learns the behavior of sgRNAs across different pools from the data. A limitation is that it might be difficult to apply RELICS to datasets that do not contain a known positive control region or that have a low number of sgRNAs overlapping positive control regions, although it is generally advisable to include positive controls in all screen designs. In addition, positive controls may not adequately represent all types of functional sequences, such as repressive or weak regulatory elements. As an alternative approach, the hyperparameters can be specified by the user. It may also be possible to develop an unsupervised learning approach where positive control labels are not provided and instead sequences with similarly-behaving sgRNAs are identified by clustering. Ideally the categories identified by such an approach would represent different types of sequences (e.g. strong regulatory elements, weak regulatory elements, silencers, non-regulatory elements).

RELICS does not currently model variation in sgRNA efficiency, variation in the strength of regulatory elements, or off-target effects. These limitations are mitigated by combining information across multiple sgRNAs, and by modeling overdispersion, which allows for the extra-variability that is introduced by factors such as off-target effects [26, 28] or sgRNA efficiency [29–31]. In addition, sgRNAs can be pre-filtered to remove those that have low predicted specificity or efficiency, as we have done in our analyses using GuideScan (see methods). Our simulations demonstrate that RELICS performs better compared to other methods, especially in the presence of weak enhancers or weak sgRNA efficiency (Fig 2C, S2 Fig, S3 Fig, S4 Fig). Nonetheless, future versions of RELICS that explicitly model these sources of variation will likely achieve even better results.

In summary, RELICS performs better than previous analysis methods under most simulation conditions and, when applied to real data, identifies both previously-validated functional sequences and new functional sequences. Thus, RELICS is an extremely useful tool for the discovery of functional sequences from CRISPR screens.

## Materials and methods

### CRSsim simulation framework

The CRSsim simulation tool is open source and available on GitHub (https://github.com/patfiaux/CRSsim). To run it, the user first specifies the screen type, which can either be a selection screen (e.g. drop out or proliferation) or an expression screen, with an arbitrary number of expression pools. Next, the user specifies the screening method (Cas9, CRISPRi, CRISPRa, dual-guide CRISPR), the number and size of true functional sequences, the spacing/density of

sgRNA target sites, and the distribution of sgRNA counts in the initial set of cells (i.e. the frequency of each sgRNA).

The probability of a cell being sorted into each pool depends on whether the sgRNA in that cell targets a functional sequence. We refer to the difference in sorting probabilities between sgRNAs that do/do not target functional sequences as the "strength of selection", $T$. The sorting of sgRNAs into the different pools is simulated by sampling from a Dirichlet multinomial distribution. The sorting probabilities of sgRNAs that do not target functional sequences are given by a vector of Dirichlet parameters $\alpha$, where the probability of being sorted into pool $j$ is $\frac{\alpha_j}{\sum_{i=1}^{J} \alpha_i}$. The sorting probabilities of an sgRNA, $n$, that overlaps a functional sequence, $k$, depends on the strength of selection, $T$, the sgRNA efficiency, $f_n$, and the strength of the functional sequence, $h_k$. Both $f_n$ and $h_k$ are proportions between 0 and 1. Guides targeting strong functional sequences, with a strength near $h_k = 1$, behave like positive control sgRNAs. Guides targeting non-functional sequences have $h_k = 0$. The simulation sets the efficiency of each sgRNA and the strength of each functional sequence by sampling from beta distributions with user-configurable shapes. Finally, the user can specify how deeply the sgRNA pools are sequenced.

The sgRNA efficiency, $f_n$, specifies the fraction of cells where the sgRNA is 'effective' and perturbs the sorting probabilities. For example, if $c$ cells contain sgRNA $n$, then $w = cf_n$ cells contain sgRNAs that are effective and they are sorted with probabilities specified by the Dirichlet vector $\alpha + h_k T$. The sorting vector for the $c - w$ cells with 'ineffective' sgRNAs, is simply $\alpha$.

Each simulation starts with an sgRNA library with a distribution of sgRNA counts. We assume the sgRNA counts follow a zero-inflated negative binomial distribution (ZINB), $y \sim ZINB(\mu, d, \varepsilon)$, where $\mu$ is the mean, $d$ is the dispersion and $\varepsilon$ is the proportion of the distribution that comes from the zero point mass. The parameters of the ZINB distribution can be specified or estimated by maximum likelihood from a provided table of sgRNA counts. After the sgRNA counts in the sgRNA library are obtained by sampling from the ZINB distribution, an input pool of cells containing sgRNAs is generated by performing multinomial sampling from the sgRNA library. We simulate the sorting of cells from the input pool into downstream pools by sampling from the Dirichlet multinomial distribution described above. Lastly, we simulate the sequencing step and obtain the count of sgRNAs in the sorted pools by drawing from a multivariate hypergeometric distribution (sampling without replacement) or a multinomial distribution (sampling with replacement). Sampling without replacement is used to simulate the use of unique molecular identifiers that allow duplicate reads to be filtered.

## Bodapati et al. simulations

We downloaded the Bodapati et al. simulation tool from GitHub (https://github.com/sbodapati/CRISPR_Benchmarking_Algorithms). We varied guide efficiency (low = 0.4 / high = 0.8), the number of guides per gene (low = 5 / high = 20) and the sequencing depth (low = 100 / medium = 200 / high = 400). We performed 15 simulations for various combinations of guide efficiency, guides per gene, and sequencing depth. This resulted in 12 scenarios and a total of 180 simulations, each with ~8400 sgRNAs and 8 functional sequences. For simulations with 5 guides per gene we simulated a total of 36 essential genes. Of those, 28 were used as training data for FS0, the remaining 8 were used as functional sequences which are to be detected. In simulations with 20 guides per gene we simulated 15 essential genes of which 7 were used as a known positive control functional sequence (FS0) and the remaining 8 were treated as unknown functional sequences. This resulted in ~150 guides as FS0 training which is similar to the datasets we analyzed in this paper. To keep the same number of guides across

the scenarios we simulated 1653 or 407 non-essential genes with either 5 guides or 20 guides per gene respectively.

Since the Bodapati et al. simulation tool was originally designed for gene-based screens we adapted the data to resemble a tiling CRISPR screen. We first randomized the order of the non-essential genes and the genes used as functional sequences while holding out the genes used for FS0. Next, we mapped all genes to genomic coordinates with guides spaced 6 bp apart (6 bp was used because that was the average tiling density of Fulco et al. 2016). All genes used for FS0 training were mapped to the middle of the tiling region to mimic the gene of interest.

## Performance assessment in simulations

We compared the performance of analysis methods for CRISPR screens using several metrics. We computed the average precision, which summarizes a precision-recall curve, using scores provided by each method. Specifically, as scores we used the functional sequence probability for RELICS, the -log10(FDR) for MAGeCK, the effect size (Beta) for CRISPR-SURF, and the Bayes Factor for BAGEL. Furthermore, we computed precision and recall using the set of regions labeled as significant by each method (BayesFactor > 5 for BAGEL, adjusted p-value < 0.05 for MAGeCK, positive significant regions output by CRISPR-SURF, and the functional sequences predicted by RELICS). Lastly, we also evaluated the fraction of all base pairs identified as significant that contained the simulated functional sequences.

The BP accuracy for the CRSSim simulations was not very high for all methods. This is due to the large area of effect (1000bp) and the small functional sequence size (50bp) used in these simulations. This area of effect is typical of CRISPRi and CRISPRa. RELICS used a genome-segment size of 100bp segments, so the best possible BP accuracy for a functional sequence that is wholly contained with a genome segment is 50%. In the case where a true FS (50bp) is split between two segments and RELICS labels correctly labels both segments as FSs, the BP accuracy is only 25%. The Bodapati simulations do not include an area of effect (i.e. the area of effect is 1bp), and RELICS used 20bp genome segments for analyzing these simulations, which allows for finer-resolution predictions and higher BP accuracy (S4 Fig).

## Running MAGeCK

MAGeCK (version 0.5.9.2) is designed to be run on sgRNAs that are grouped into functional units (i.e. genes). To run MAGeCK, we therefore grouped sgRNAs into non-overlapping genome windows containing 10 sgRNAs per window (note that the GeCKO v2 sgRNA library that MAGeCK is commonly applied to uses 6 sgRNAs per gene). To run MAGeCK we used the following command line:

```
mageck test -k MAGeCK_Input.txt -t 0,1 -c 2,3 -n mageckOut
```
where the MAGeCK_Input.txt file contains the observed counts for each sgRNA.

## Running BAGEL

BAGEL (downloaded from sourceforge on 12/12/2019) is designed to be run on sgRNAs that are grouped into functional units (i.e. genes). To run BAGEL we grouped sgRNAs using the same 10-sgRNA genomic windows that we used for MAGeCK. Since BAGEL takes log2 fold change as input, we calculated log2 fold change from the sgRNA counts. BAGEL is a super-vised method and requires positive and negative controls as training. We provided a set of pos-itive and negative sgRNAs to be used as such. For the simulations, the positive sgRNAs were taken from the simulated 'positive control region'. For the experimental datasets, we provided the same positive controls that RELICS used for training. As negative sgRNAs we used the first

few 10-sgRNA windows which corresponded to the background and selected the same number of windows as was used as positive controls. To run BAGEL we used the following command:

```
python BAGEL.py -i BAGEL_Input.txt -o BAGEL_out.bf -e Input_po-
sTrain.txt -n Input_negTrain.txt -c 1,2
```

## Running CRISPR-SURF

CRISPR-SURF (downloaded 02/13/2019 from GitHub) was run using the procedure specified in the supplemental materials of Hsu et al. 2018. CRISPR-SURF analyzes count data in two steps. The first step filters out guides of low quality and computes log2 fold change (command 'SURF_count'). The second step performs the deconvolution (command 'SURF_deconvolution') to identify functional sequences. During the first step, all guides which have a count less than 50 in any of the pools are removed. While this works with data sets that have been sequenced deeply, this can become an issue when the sequencing depth is not as high. For this reason, to run CRISPR-SURF on the simulated datasets, we set the guide count filter to 15 instead of 50. For the experimental datasets we used the default guide count filter of 50. CRISPR-SURF accepts positive control as input, which can improve the output. We provided CRISPR-SURF with the same positive controls that RELICS used for training. The SURF_-count command was run as follows:

```
docker run -v $Path_to_file/:$Path_to_file -w $Path_to_file
pinellolab/crisprsurf SURF_count -f CRISPR_SURF_count.csv -nuclease
cas9 -pert crispri
```

The SURF_deconvolution command was run as follows:

```
docker run -v $Path_to_file/:$Path_to_file -w $Path_to_file
pinellolab/crisprsurf SURF_deconvolution -f CRISPR_SURF_Input.csv
-pert crispri
```

## Running RELICS

To run RELICS on the CRSsim data we used the default parameters for analyzing a CRISPRi screen. For analyzing the Bodapati et al. simulation data we used a genome segment size of 20bp in addition to the default parameters for a Cas9 screen. To run RELICS on experimental datasets we used the default parameters for a CRISPRi/CRISPRa screen which has an sgRNA area of effect of 400bp (+/-200bp around each target site). The promoter regions of the genes of interest were used as positive control regions. The same positive control regions were used for CRISPR-SURF and BAGEL.

## CRISPRa screen data

We obtained data from published CRISPRa screens for *IL2RA* and *CD69* gene expression in Jurkat T cells [9]. We downloaded the sgRNA counts from the paper supplement and converted them to the RELICS input format. The promoter regions were used as positive control regions for RELICS and CRISPR-SURF and were defined as the transcription start site (TSS) +/- 1kb. RELICS jointly analyzed all pools for the analysis. For MAGeCK and CRISPR-SURF, the input pool was used as control pool and the high-expression pool was used as the treatment pool. RELICS and CRISPR-SURF used the same positive control sgRNAs.

## CRISPRi screen data

We obtained data from published CRISPRi proliferation screens for *GATA1* and *MYC* in K562 cells [3]. We downloaded the sgRNA counts from the paper supplement and converted them to the RELICS input format. For detecting *MYC* regulatory elements, all sgRNAs labelled as '*MYC* Tiling' sgRNAs as well as 'Protein Coding Gene Promoters' sgRNAs were used. For

detecting *GATA1* regulatory elements, all sgRNAs labelled as '*GATA1* Tiling' as well as 'Protein Coding Gene Promoters' were used. For MAGeCK and CRISPR-SURF, the pool at T0 (before) was used as the treatment pool and the pool at T14 (after) was used as the control pool in order to look for enrichment instead of depletion. RELICS and CRISPR-SURF used the same positive control sgRNAs.

### Filtering guides with GuideScan

We filtered all guides from both the CRISPRi and CRISPRa experimental screens using Guide-Scan [32]. All possible guides for the regions targeted in each screen were obtained from the online version of the GuideScan tool (http://www.guidescan.com/, used 05/08/2020). Guide-Scan eliminates all guides with either a perfect match or a 1bp mismatch at any other site in the genome. All remaining guides receive a specificity score. We used the specificity score cut-off of 0.2 recommended by Tycko et al. [33]. All remaining guide sequences were then matched to the guide sequences from the studies described above. Filtering sgRNAs with GuideScan reduced the total number of guides in each study substantially (S1 Table), however removal of these sgRNAs did not substantially affect the results. Results for *CD69* without GuideScan filtering are shown in S9 Fig.

### H3K27ac data

Jurkat H3K27ac data from Mansour et al. [34] was downloaded from GEO (GSM1296384). The reads were aligned with BWA-MEM [35] using default parameters and filtered for duplicates and low mapping quality (MAPQ < 30) using samtools [36]. The reads were then converted to reads per kilobase per million in 200bp bins using deepTools2 [37]. ENCODE H3K27ac ChIP-seq data for the K562 cell line was downloaded in bedgraph format from the UCSC genome browser.

### CRISPRa validation experiments

We used FlashFry [31] to design two sgRNAs per target site. FlashFry calculates guide efficiency and off-target effects for each guide by reporting the Doench 2014 [38], Doench 2016 [26], and Hsu 2013 scores [28]. For each of the sites we selected guides with low estimated off-target effects and high estimated efficiency relative to the other possible guides in the region (S2 Table). The designed sgRNAs were cloned into pCRISPRiaV2 plasmids (Addgene #84832). Lentiviruses carrying the sgRNAs were generated and transduced into Jurkat cells expressing dCas9-VP64 [9], which were obtained from the Berkeley Cell Culture Facility. 7–10 days after transduction, cells from different treatments were stained with PE anti-human CD69 antibody (Biolegend #310906) and the expression of CD69 was measured by flow cytometry using a BD FACSCanto II system.

### CRISPRi validation experiments

K562 cells were transduced with dCas9-KRAB-GFP lentivirus (Addgene #71237) carrying different sgRNAs (S3 Table, S4 Table). The percentage of GFP positive cells was recorded by FACS 3 days after transduction (D0) and again after an additional 6 (D6) and 14 (D14) days of culture. Two biological replicates (from separate cultures) were performed at each time point.

### RELICS model setup and organization

A description of the variables for the RELICS model is provided in S3 Table. RELICS divides the screened genome region into *M* small non-overlapping genome segments indexed as

$m = 1,2,\ldots,M$. In practice we have found a segment size of 100bp to work well, and we use this as the default. As input, RELICS takes a table of observed counts for $N$ sgRNAs (indexed as $n = 1,2,\ldots,N$) across $J$ pools. We represent these counts as an $N\times J$ matrix, $y$, and use $y_n$ to denote the vector of observed counts for sgRNA $n$ across the $J$ pools. Let $g(n)$ be a function that maps sgRNAs to associated 'overlapping' genome segments. I.e. $g(n)$ returns the set of genome segments that are overlapped by sgRNA $n$. RELICS does not use non-targeting sgRNAs.

## RELICS sgRNA count model

RELICS assumes there are $K$ functional genome sequences, indexed as $k = 1,2,\ldots,K$ and that each functional sequence affects the sorting probabilities of overlapping sgRNAs. Each functional sequence has a length, $l_k$, which is the number of genome segments that it spans. The maximum length of a functional sequence is set to $L$, so that $l_k \in \{1,\ldots,L\}$. By default, $L = 10$.

We describe the observed counts for an sgRNA across pools with a Dirichlet multinomial distribution and refer to the Dirichlet portion of the distribution as the sorting probability distribution. The sgRNAs that overlap a functional genome sequence have one sorting probability distribution, and the sgRNAs that do not overlap a functional genome sequence have another. Each Dirichlet distribution has $J$ shape parameters, which define the dispersion and the expected proportion of counts in each pool. The vectors of shape parameters for the Dirichlet distributions are denoted $\alpha_1$ and $\alpha_0$ for sgRNAs that do and do not overlap a functional sequence. Then:

$$y_n | s_n \sim \text{Multinomial}(s_n)$$

$$s_n | r_n = 0 \sim \text{Dirichlet}(\alpha_0)$$

$$s_n | r_n > 0 \sim \text{Dirichlet}(\alpha_1)$$

where $r_n$ is the total number of genome segments containing functional sequences that are overlapped by sgRNA $n$.

## RELICS functional sequence configurations

We define a configuration to be the positions and lengths of all of the functional sequences. We specify a configuration with a matrix, $\delta$, of dimension $K\times M$, where an element, $\delta_{k,m}$, is 1 if genome segment $m$ contains functional sequence $k$, and is 0 otherwise. We call a single row vector of the configuration matrix, $\delta_k$, the placement of a functional sequence. In other words, a configuration is a collection of functional sequence placements.

We want to estimate the probability, $p_m$, that a given genome segment, $m$, contains a functional sequence. To compute $p_m$ we could sum the posterior probabilities of all configurations that have a functional sequence in genome segment $m$. However, exact calculation of $p_m$ is intractable because the likelihoods of all possible configurations must be computed. For example, even in a simple case where there are $M = 10,000$ genome segments and $K = 5$ regulatory sequences of length $L = 1$, the number of possible $\delta$ configurations is $\binom{10,000}{5} = 8.3 \times 10^{17}$.

To overcome this problem, we developed an approximate inference algorithm known as Iterative Bayesian Stepwise Selection (IBSS) [25]. Our version of the IBSS algorithm includes extensions that are specific to RELICS including allowing for functional sequences of variable length and the use of non-normal count data with a Dirichlet multinomial error distribution. Our IBSS algorithm performs stepwise placement of a single functional sequence at a time, while accounting for the (uncertain) placements of all of the other functional sequences.

To implement the IBSS algorithm, and to account for uncertainty in functional sequence placements, we introduce a functional sequence probability matrix, $\boldsymbol{\pi}$. This is like the $\boldsymbol{\delta}$ matrix, but rather than binary values, it contains probabilities. Specifically, each element is the probability that a genome segment contains a specific functional sequence: $\pi_{k,m} = \Pr(\delta_{k,m} = 1)$.

To allow the positions of known functional sequences (positive controls) to be specified, we add an additional row to the functional sequence probability matrix, which we index as row 0 and denote $\boldsymbol{\pi_0}$. The probability that a genome segment contains *any* functional sequence is then: $p_m = \sum_{k=0}^{K} \pi_{k,m}$.

The number of genome segments that are overlapped by sgRNA $n$ and contain a functional sequence follows a Poisson binomial distribution. In other words, the Poisson binomial is used to calculate the probability that an sgRNA overlaps $r_n$ genome segments containing functional sequences:

$$r_n \sim \mathrm{PoissonBinomial}\left(\boldsymbol{p}_{g(n)}\right)$$

where $\boldsymbol{p}_{g(n)}$ is the vector of probabilities for all genome segments associated with sgRNA $n$.

## RELICS IBSS algorithm

The following is a description of the IBSS algorithm that is used to estimate the functional sequence probability matrix $\boldsymbol{\pi}$.

<u>Initialize:</u>

- Set $K$ to number of functional sequences.

- Set known functional sequences (positive controls) to 1.0 in row $\boldsymbol{\pi_0}$.

- Set all other elements of $\boldsymbol{\pi}$ to 0.0.

    While not converged:

- Estimate sgRNA sorting hyperparameters ($\alpha_0$, $\alpha_1$) by maximum likelihood.

    # Estimate configuration probabilities
- For $k$ in 1...$K$:

    ○ Set elements of row $\boldsymbol{\pi_k}$ to 0.

    ○ Compute $\boldsymbol{p}$, the probability each genome segment contains one of the other functional sequences

    ○ Set elements of $\boldsymbol{\pi_k}$ by calculating posterior probability of every possible placement of functional sequence $k$, conditional on $\boldsymbol{p}$.

The algorithm is considered to be converged when the maximum absolute difference in the updated functional sequence probability matrix ($\max(\mathrm{abs}(\boldsymbol{\pi'} - \boldsymbol{\pi}))$) less than a defined threshold.

## RELICS posterior probabilities of functional sequence placements

To compute the posterior probability of each functional sequence placement, we first compute the likelihood of all possible placements of functional sequence $k$, taking into account the placements of all of the other functional sequences. We set the elements of $\boldsymbol{\pi_k}$ to 0 and also set all elements of $\boldsymbol{\delta_k}$ to 0, except those corresponding to the functional sequence placement

being considered, which are set to 1. We denote a specific placement as $\boldsymbol{\delta_k}^{<m,l>}$, where $m$ is the genome segment containing the start of the functional sequence, and $l$ is the length of the functional sequence. That is, $\boldsymbol{\delta_k}^{<m,l>}$, is a vector of 0s except for elements $m.(m+l-1)$, which are set to 1. We can compute the probability that a given genome segment contains a functional sequence as $p_m = \delta_{k,m} + (1 - \delta_{k,m}) \sum_{k=0}^{K} \pi_{k,m}$. The likelihood of a specific functional sequence placement is then:

$$\mathcal{L}\left(\boldsymbol{\delta_k}^{<m,l>}|\boldsymbol{y}, \boldsymbol{\alpha}_1, \boldsymbol{\alpha}_0, \boldsymbol{\pi}\right) = \prod_n \left[\Pr_{PB}\left(r_n > 0|\boldsymbol{p}_{g(n)}\right)\Pr_{DMN}(\boldsymbol{y}_n|\boldsymbol{\alpha}_1) + \Pr_{PB}\left(r_n = 0|\boldsymbol{p}_{g(n)}\right)\Pr_{DMN}(\boldsymbol{y}_n|\boldsymbol{\alpha}_0)\right]$$

where the product is over all sgRNAs with observed counts, $\Pr_{DMN}(\boldsymbol{y}|\boldsymbol{\alpha})$ is the probability of the observed counts (computed using the Dirichlet multinomial distribution), and $\Pr_{PB}(r|\boldsymbol{p})$ is the probability an sgRNA has $r$ overlapping functional sequences (computed using the Poisson binomial distribution).

The posterior probability of a specific functional sequence placement is:

$$PP_{\boldsymbol{\delta_k}^{<m,l>}} = \frac{\Pr\left(\boldsymbol{\delta_k}^{<m,l>}\right)\mathcal{L}\left(\boldsymbol{\delta_k}^{<m,l>}|\boldsymbol{y}, \boldsymbol{\alpha}_1, \boldsymbol{\alpha}_0, \boldsymbol{\pi}\right)}{\sum_{i,j}\Pr\left(\boldsymbol{\delta_k}^{<i,j>}\right)\mathcal{L}\left(\boldsymbol{\delta_k}^{<i,j>}|\boldsymbol{y}, \boldsymbol{\alpha}_1, \boldsymbol{\alpha}_0, \boldsymbol{\pi}\right)}$$

where the denominator is the sum over all possible placements of this functional sequence and $\Pr(\boldsymbol{\delta_k}^{<m,l>})$ is the prior probability of a specific functional sequence placement.

We can compute the posterior probability that a genome segment contains functional sequence $k$, by summing the probabilities from all of the possible placements overlapping the segment. We use these posteriors to set the elements of $\boldsymbol{\pi_k}$:

$$\pi_{k,m} = \sum_{l=1}^{L} \sum_{i=m-l+1}^{m} PP_{\boldsymbol{\delta_k}^{<i,l>}}$$

## RELICS prior probabilities of functional sequence placements

As prior probabilities for each functional sequence placement, we use a weighting that favors shorter functional sequences. Specifically, we use a geometric distribution, truncated at a maximum of $L$, to weight each possible length:

$$w(l) = \frac{(1-\lambda)^{l-1}\lambda}{\sum_{i=1}^{L}(1-\lambda)^{i-1}\lambda}$$

where $\lambda$ is a constant between 0 and 1 that controls the weighting. We make the prior uniform for all placements with the same value of $l$.

## RELICS empirical estimation of hyperparameters

The RELICS model has hyperparameters, $\boldsymbol{\alpha_0}$ and $\boldsymbol{\alpha_1}$, which control the sorting probabilities and dispersion of sgRNA counts across pools. RELICS performs maximum likelihood estimation (MLE) of these parameters each iteration of the IBSS algorithm. This estimation is performed using the full dataset of sgRNA counts, and keeping the functional sequence probabilities fixed to their current estimates, $\hat{\boldsymbol{\pi}}$:

$$\alpha_0, \alpha_1 = \text{argmax}_{\alpha_0,\alpha_1} \mathcal{L}(\boldsymbol{\alpha}_1, \boldsymbol{\alpha}_0|\boldsymbol{y}, \hat{\boldsymbol{\pi}})$$

We perform MLE by numerical optimization using the L-BFGS-B algorithm [39].

## RELICS choice of the number of functional sequences ($K$)

RELICS estimates the number of functional sequences by first setting $K$ to 1, and using the IBSS algorithm to estimate the functional sequence probability matrix, $\pi$. $K$ is then increased by one and the process is repeated. Each iteration we compute the pairwise Pearson correlations between all rows of $\pi$. Initially the pairwise correlations are very low, however once the value of $K$ is too high, the functional sequences placed by the algorithm begin to overlap causing positive correlations between pairs of functional sequence probabilities. In both simulations and real data, we have found it practical to set $K$ to the largest value that yields a maximum pairwise correlation of less than 0.1. Fig 2, S5 Fig, and S7 Fig show how the maximum pairwise correlations increase dramatically once $K$ exceeds the number of distinct functional sequences that can be identified from the data.

## RELICS input format

RELICS takes a text-based.csv file as input (for examples see https://github.com/patfiaux/RELICS). The file contains the sgRNA information as well as the observed sgRNA counts in different pools. We have made the public datasets we analyzed available in this format at https://figshare.com/projects/RELICS_2_data/74376. Alternatively, RELICS can also take in two.csv files, where one file contains the sgRNA information and the other file contains the observed sgRNA counts in different pools.

## Availability of data and materials

RELICS is open source and available on GitHub (https://github.com/patfiaux/RELICS). All data were obtained from published papers or simulated as described above. The formatted data that we used for analyses can be downloaded from: https://figshare.com/projects/RELICS_2_data/74376.

# Supporting information

**S1 Fig.** CRISPR screen simulation framework (A) Experimental workflow. sgRNAs are introduced into cells and only cells which receive sgRNAs are retained. These cells are then either sorted based on gene expression, or placed under selective pressure (e.g. for survival or proliferation). sgRNA counts and guide ranks before and after selection are shown. (B) In our simulations, we generate an initial sgRNA count distribution and mimic the experimental steps using several different sampling procedures. The simulated sgRNA counts for before and after simulated selection are shown for comparison with the experimental data. (C) Quantile-quantile plots of simulated vs experimental data, before and after selection. (D) Combinations of parameters used for the simulations. We simulated two different types of screens with two different selection strengths, 3 different guide efficiency distributions, and two sequencing depths. For each scenario, we simulated 30 data sets.
(TIF)

**S2 Fig. Performance evaluation of analysis methods on simulated data for a selection screen with 2 pools.** (A) Boxplots of average precision for different simulation parameters. (B) Boxplots of precision, the fraction of significant regions that contain a simulated enhancer. (C) Boxplots of recall, the number of simulated enhancers detected (out of 8) amongst the significant regions identified. (D) Boxplots of base pair (BP) accuracy, the fraction of base pairs in significant regions that overlap a simulated enhancer. The hinges of the boxplots correspond to the first and third quartiles, the center lines are the medians, and the whiskers extend to the

furthest datapoints that are within 1.5x the interquartile range from the hinge.
(TIF)

**S3 Fig. Performance evaluation of analysis methods on simulated data for a gene expression (FACS) screen with 4 pools.** (A) Boxplots of average precision for different simulation parameters. (B) Boxplots of precision, the fraction of significant regions that contain a simulated enhancer. (C) Boxplots of recall, the number of simulated enhancers detected (out of 8) amongst the significant regions identified. (D) Boxplots of base pair (BP) accuracy, the fraction of base pairs in significant regions that overlap a simulated enhancer. The hinges of the boxplots correspond to the first and third quartiles, the center lines are the medians, and the whiskers extend to the furthest datapoints that are within 1.5x the interquartile range from the hinge.
(TIF)

**S4 Fig. Performance evaluation of analysis methods on data simulated using the simulation tool by Bodapati et al.** The simulated data is for a selection screen with 2 pools. (A) Boxplots of average precision for different simulation parameters. (B) Boxplots of precision, the fraction of significant regions that contain a simulated enhancer. (C) Boxplots of recall, the number of simulated enhancers detected (out of 8) amongst the significant regions identified. (D) Boxplots of base pair (BP) accuracy, the fraction of base pairs in significant regions that overlap a simulated enhancer. The hinges of the boxplots correspond to the first and third quartiles, the center lines are the medians, and the whiskers extend to the furthest datapoints that are within 1.5x the interquartile range from the hinge. The simulation details are described in the methods. In 117/180 of the simulations RELICS had the highest AP. RELICS also had the highest precision (157/180), recall (163/180) and BP accuracy (102/180).
(TIF)

**S5 Fig. Log-likelihood progression and pairwise correlations between functional sequence (FS) probabilities as the number of functional sequences ($K$) is increased.** (A,C,E) Log likelihood as the number of FSs increases for *CD69*, *MYC* and *GATA1*. (B,D,F) Pairwise FS placement probability correlations for *CD69*, *MYC* and *GATA1*. The red dashed lines indicate the highest pairwise correlation across all FSs. The hinges of the boxplots in (B), (D) and (F) correspond to the first and third quartiles, the center lines are the medians, and the whiskers extend to the furthest datapoints that are within 1.5x the interquartile range from the hinge.
(TIF)

**S6 Fig. Experimental validation experiments for *CD69*.** (A) Analysis results for the CD69 screen by RELICS, CRISPR-SURF, MAGeCK and log2 fold change. The RELICS probabilities for each FS are collapsed into a single track. An H3K27ac ChIP-seq track for Jurkat cells is included. The locations of sequences chosen for experimental validation are indicated by colored lines. (B) Results from validation experiments. Lentiviral vectors encoding sgRNAs and dCas9:VP64 were co-transduced into Jurkat cells, and the expression of CD69 protein was quantified by flow cytometry. Non-targeting sgRNAs were used as a negative control. Targeting sgRNAs were chosen for their specificity and high predicted efficiency (relative to other possible sgRNAs in the region) and in some cases are adjacent to the predicted FS rather than within the FS.
(TIF)

**S7 Fig. Analysis of a CRISPRa screen for *IL2RA* expression published by Simeonov et al. 2017.** (A) Log-likelihood progression. With the addition of each functional sequence (FS), the model log-likelihood is recorded. (B) The pairwise correlations between functional sequence

probabilities as a function of the number of FSs. The red dashed line indicates the highest pair-wise correlation between all pairs of FSs. The hinges of the boxplots correspond to the first and third quartiles, the center lines are the medians, and the whiskers extend to the furthest data-points that are within 1.5x the interquartile range from the hinge. (C) Output of RELICS and another analysis methods. Each FS predicted by RELICS is given a different color and the labels are arranged by genomic position.
(TIF)

**S8 Fig. Analysis of a published CRISPR inhibition (CRISPRi) cellular proliferation screen around *GATA1* and experimental validation of sequence that is not predicted by RELICS.**
(A) Results from RELICS and other analysis methods. RELICS detects 5 functional sequences (FS1-5). FS1 and FS2 have previously been validated; FS3 and FS4 fall within GATA1. sgRNA target sites for validation experiments are indicated with a red circle (GATA1 promoter) and grey and black triangles (*GLOD5*). (B) Results from validation experiments (2 replicates) using sgRNAs targeting sites indicated in panel A. Each validation experiment is a cellular prolifera-tion assay, in which the percent of GFP-positive cells (i.e. those that received the sgRNA) are measured at day 0, day 6 and day 14. While targeting the *GATA1* promoter greatly reduces proliferation, targeting the *GLOD5* region does not change proliferation compared to a nega-tive control sgRNA, which targets a non-functional 'safe harbor' region on chromosome 8.
(TIF)

**S9 Fig. Analysis results for a CD69 CRISPRa screen without GuideScan filtering and after downsampling sgRNAs or sgRNA counts.** sgRNA counts were downsampled to 25% of all counts or sgRNA density was downsampled to 25%. Dashed blocks group results from differ-ent analysis methods together (RELICS, CRISPR-SURF, MAGeCK, log2 Fold Change).
(TIF)

**S1 Table. Number of sgRNAs present in each data set before and after filtering with Guide-Scan.**
(XLSX)

**S2 Table. *CD69* CRISPRa validation sgRNAs. TSS sgRNAs target the transcription start site of *CD69*.** FSs, and 'Neg. Target' sgRNAs correspond to labels in Fig 4D. NTC is a non-tar-geting control sgRNA.
(XLSX)

**S3 Table. *MYC and GATA1* CRISPRi validation sgRNAs.** Guide sequences are given for each target. The positive control targets a known functional sequence (e2) identified by Fulco et al. As negative controls a dCas9:KRAB only or an sgRNA targeting a non-functional safe harbor region on chromosome 8 were used.
(XLSX)

**S4 Table. *GATA1* CRISPRi validation sgRNAs.** Coordinates are given in both hg38 and hg19 genome assemblies. NS is a negative control sgRNA targeting a safe harbor region on chromo-some 8. GATA1_TSS sgRNA targets the transcription start site of *GATA1*. The Glod5-1 / Glod5-2 sgRNAs target the putative functional sequence near *GLOD5* identified by Fulco et al. 2016.
(XLSX)

**S5 Table. Description of variables in the RELICS model.**
(DOCX)

## Acknowledgments

We thank Yarui Diao, Rongxin Fang, Ye Zheng, Zhi Liu, and members of the McVicker lab for helpful discussions about analysis methods for CRISPR regulatory screens; Arko Sen for guidance in obtaining and processing public datasets; Bing Ren, for his laboratory's assistance with CRISPRi validation experiments; and Jessica Zhou and Arya Massarat for testing RELICS and CRSsim.

## Author Contributions

**Conceptualization:** Patrick C. Fiaux, Hsiuyi V. Chen, Graham McVicker.

**Data curation:** Patrick C. Fiaux.

**Formal analysis:** Patrick C. Fiaux.

**Funding acquisition:** Graham McVicker.

**Investigation:** Patrick C. Fiaux, Hsiuyi V. Chen, Poshen B. Chen, Aaron R. Chen, Graham McVicker.

**Methodology:** Patrick C. Fiaux, Hsiuyi V. Chen, Graham McVicker.

**Project administration:** Graham McVicker.

**Software:** Patrick C. Fiaux.

**Supervision:** Graham McVicker.

**Validation:** Hsiuyi V. Chen, Poshen B. Chen, Aaron R. Chen.

**Visualization:** Patrick C. Fiaux.

**Writing – original draft:** Patrick C. Fiaux, Graham McVicker.

**Writing – review & editing:** Patrick C. Fiaux, Hsiuyi V. Chen, Graham McVicker.

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
