## [Decision Letter · Decision Letter 0]

4 May 2020

Dear Dr. McVicker,

Thank you very much for submitting your manuscript "Discovering functional sequences with RELICS, an analysis method for CRISPR screens" for consideration at PLOS Computational Biology.

As with all papers reviewed by the journal, your manuscript was reviewed by members of the editorial board and by several independent reviewers. In light of the reviews (below this email), we would like to invite the resubmission of a significantly-revised version that takes into account the reviewers' comments.

We cannot make any decision about publication until we have seen the revised manuscript and your response to the reviewers' comments. Your revised manuscript is also likely to be sent to reviewers for further evaluation.

Sincerely,

Ilya Ioshikhes

Associate Editor

PLOS Computational Biology

Jian Ma

Deputy Editor

PLOS Computational Biology

Reviewer's Responses to Questions

**Comments to the Authors:**

**Reviewer #1: **The study entitled "Discovering functional sequences with RELICs, an analysis method for CRISPR screens" by Fiaux et al describes a novel method for analyzing pooled genetic screens. Overall, this study is clearly written and uses an innovative approach for screen analytics, which is distinct from previously described methods (BAGEL, MAGECK). The use of simulated and 'real' data is effective and strongly supports the power of this approach. In summary, I believe this study describes a useful tool that will be of broad utility to the field.

**Reviewer #2: **In this manuscript, Fiaux et al. describe Regulatory Element Location Identification in CRISPR Screens (RELICS), a computational method to identify functional regulatory sequences from CRISPR tiling screens using sgRNA counts from multiple pools. To evaluate the performance of their method, the authors also developed a CRISPR simulation framework, CRSsim. They generated a total of 720 simulations using this framework by varying the strength of the guide, sequencing depth and the strength of selection. RELICS was tested on these simulated datasets and then compared to the predictions from BAGEL, MAGeCK and CRISPR-SURF. The authors then applied their method to existing CRISPRa/i tiling screens to show that RELICS predicts functional sequences predicted by other existing algorithms and also novel ones. They validated RELICS predicted functional sequences by using CRISPRa to overexpress CD69 assayed by flow cytometry.

Overall, RELICS could potentially help in discovering functional sequences using CRISPR screens but this manuscript needs to be supported with more evidence that this method is better than existing tools. The novelty of this method is not very clear in the current state of the manuscript. The lack of “ground truth” in most of these cases could be a problem. The authors clearly note in the discussion that prior knowledge of a positive control region is necessary to run this analysis and might be difficult to obtain. The existing tools used for comparison also do not seem relevant as elaborated below.

Major points

The authors mention that the simulation framework CRSsim generates “realistic datasets where the ground truth is known”. A figure comparing the data generated by this framework to some existing CRISPR datasets would strengthen this manuscript.Base pair accuracy (BP) is defined as the fraction of significant bases identified which are true functional sequences. How are “true” functional sequences identified here? Also, in Figure 2, RELICS does predict functional sequences with higher BP accuracy than other methods especially for a guide with lower efficiency and weak selection strength but the BP accuracy does not get better for stronger guides or stronger selection strengths. Why is that the case?The authors use four other methods to compare RELICS to. The log-fold change is just a simple transformation of the data and not really a “method”. Likewise, BAGEL and MAGeCK are approaches to find essential genes, not identify functional elements. These comparisons should be excluded and the only relevant comparison to CRISPR-SURF should be included in this manuscript.The authors also show that their method is better than BAGEL, MAGeCK and CRISPR-SURF using precision-recall AUC and BP accuracy on the simulated datasets. These metrics should also be used to demonstrate that RELICS is better than the other methods using the published data sets.Only a single guide is used to validate the “negative sequence” prediction that was made by MAGeCK and is in a significant portion predicted by CRISPR-SURF but not RELICS. The Rule Set 2 score of this guide is 0.39 which makes it an inefficient guide. The authors do note in the legend of Figure 4 that “Targeting sgRNAs were chosen for their high predicted efficiency”. Based on what criteria were the targeting sgRNAs picked? Also, when a single guide is used for validation, it is difficult to determine if CD69 activation was not observed due to the inefficiency of the guide or the region did not indeed have a functional sequence. The authors do validate most other regions with 2 sgRNAs but why was this region only targeted with one guide? To support their claim of RELICS identifying new regions, the authors should validate the 5 regions identified around the MYC locus that have not been reported previously. This would help in proving the novelty of this method.RELICS was easy to use and the instructions were very clear on GitHub. Well documented tools are hard to find and the authors have done a very good job on explaining the various parameters that can be used to run this tool.Bodapati et al., 2020 describes a simulation framework for CRISPR screens and the use of similar hyperparameters as described in this paper. It might be useful to compare CRSsim to their framework and also apply RELICS to datasets simulated by their framework.

Minor points

Typo in Line 152, it should be “functional sequence probabilities”.In line 157, the authors mention that RELICS predictions on the simulated datasets were compared to “log-fold change”, but this comparison has neither been demonstrated in the main or supplementary figures nor has been further discussed in the text.In line 190, Figure 3(a, b) are referenced but do not directly correspond to the text. Figure 3(c, d) are referenced after Figure 4. It might be helpful to re-order the figures to go with the flow of the text.In Figure 5b., indicate what the two lines for GLOD5 sgRNA1, GLOD5 sgRNA2 and Negative control indicate? Are they replicates?Supplementary Figure 3 & 4 indicate the method used as “RELICS 2” but other figures say “RELICS ''. Please explain the difference between these methods.Please include why and how sgRNAs were filtered to run CRISPR-SURF on the experimental datasets?Another analysis that might be worth adding would be to filter out promiscuous guides from the tiling libraries as shown by Tycko et al., 2019 and re-assessing the performance of RELICS.

**Reviewer #3: **The paper by Fiaux et al. introduces a Bayesian hierarchical model, which the authors termed RELICS, designed to help identifying functional sequences using CRISPR screens. The model performs best when offered data that contain positive controls (known functional sequences) and as such uses a semi-supervised approach. To test the performance of their model, the authors also developed an open source simulation tool, CRSsim, which should be useful for cross-validation of performances of different models. The RELICS method seems interesting and may embody a considerable advance as it outperformed existing models in the analyses shown. The manuscript is well-written and the data are aesthetically well present.

Major point:

The RELICS methodology is not well explained and confusingly presented. Although most of the information can be found in the manuscript, most readers will miss it because it takes a while to find. In some sections the presentation is a bit too technical and as such makes for a tedious read. To make this manuscript more accessible to a broader authorship, the authors are encouraged to walk the reader through the key design steps of RELICS in plain language, to reduce jargon and where needed explain terminology on first use. In its current form, readers will also find it difficult to fully appreciate the strengths of the methodology and to independently validate that comparisons to existing models were undertaken in a way that did not favor RELICS.

Minor points:

Figure 1a: The ‘area of effect’ depictions in this schematic are indicating identical spans for each guide. In reality, guides that target overlapping functional sequences would be expected to often have the boundaries of their ‘areas of effect’ coincide.

Figure 1b and c: The authors chose to depict guides in different pools with the same colors, which I presume does not indicate that indeed the guides used in the different pools were identical? This is confusing and should be changed unless I misunderstood the methodology.

P18, Lines 335 -369: It seems that positive controls (i.e., known functional sequences) were specified only for RELICS, not any of the other platforms it performance was compared against. Since the authors mentioned that RELICS may be difficult to apply to datasets without an identified positive control (line 265), would the lack of a positive control have disadvantaged the performance of the other platforms, and if so, to which extent would this have compromised their performance when compared with RELICS?

P23, Lines 487-488: Note that the font is changing in unexpected ways in this section.

Overall, this reviewer is excited about this paper but strongly advises the authors to change the presentation such that the key design features and steps distinguishing RELICS from existing methodologies are explained early on.

**Have all data underlying the figures and results presented in the manuscript been provided?**

Reviewer #1: Yes

Reviewer #2: None

Reviewer #3: Yes

PLOS authors have the option to publish the peer review history of their article (what does this mean?). If published, this will include your full peer review and any attached files.

Reviewer #1: No

Reviewer #2: No

Reviewer #3: No
---

## [Decision Letter · Decision Letter 1]

25 Jul 2020

Dear Dr. McVicker,

We are pleased to inform you that your manuscript 'Discovering functional sequences with RELICS, an analysis method for CRISPR screens' has been provisionally accepted for publication in PLOS Computational Biology.

Best regards,

Ilya Ioshikhes

Associate Editor

PLOS Computational Biology

Jian Ma

Deputy Editor

PLOS Computational Biology

Reviewer's Responses to Questions

**Comments to the Authors:**

Reviewer #2: Fiaux et al. have addressed my concerns regarding this manuscript. I appreciate that they took the time to assess the existing simulation framework. The validation of novel functional sequences identified by RELICS adds a lot of value to this manuscript. I believe that this tool will be useful to the community for analysis of non-coding CRISPR screens.

Reviewer #3: The authors are commended for their sincere efforts to address reviewer comments. The revised manuscript is considerably improved and should be of interest to the pertinent readership in this growing field of study.

**Have all data underlying the figures and results presented in the manuscript been provided?**

Reviewer #2: Yes

Reviewer #3: Yes

PLOS authors have the option to publish the peer review history of their article (what does this mean?). If published, this will include your full peer review and any attached files.

Reviewer #2: No

Reviewer #3: No

---

## [Editor Report · Acceptance letter]

10 Sep 2020

PCOMPBIOL-D-20-00463R1 

Discovering functional sequences with RELICS, an analysis method for CRISPR screens

Dear Dr McVicker,

I am pleased to inform you that your manuscript has been formally accepted for publication in PLOS Computational Biology. Your manuscript is now with our production department and you will be notified of the publication date in due course.

With kind regards,

Laura Mallard
